# Cheese Whey Protein and Blueberry Juice Mixed Fermentation Enhance the Freeze-Resistance of Lactic Acid Bacteria in the Freeze-Drying Process

**DOI:** 10.3390/foods13142260

**Published:** 2024-07-17

**Authors:** Yuxian Wang, Xian Liu, Yufeng Shao, Yaozu Guo, Ruixia Gu, Wenqiong Wang

**Affiliations:** College of Food Science and Engineering, Yangzhou University, Yangzhou 225127, China; wangyuxian0708@163.com (Y.W.); liuqian7734@126.com (X.L.); shaoyufeng20000903@163.com (Y.S.); gyz15240477626@163.com (Y.G.); guruixia1963@163.com (R.G.)

**Keywords:** blueberry–whey protein mixed fermentation, freeze-drying, lactic acid bacteria, process optimization

## Abstract

The effects of MRS, whey protein and blueberry alone, and mixed fermentation on the survival rate of lactic acid bacteria under various freeze-drying conditions were investigated. The surface structure of the freeze-dried powders was also investigated to explore the anti-freezing protection mechanism of mixed whey protein and blueberry fermentation on the bacteria. It was found that the mixed fermentation medium of blueberry and whey protein has a protective effect on the freeze-drying bacteria and is better than the traditional MRS and whey protein medium. The optimal concentration of blueberry juice addition was 9%. The survival rate of the pre-freezing temperature at −80 °C was higher than at −20 °C after the pre-freezing and freeze-drying processes. The freeze-drying thickness of 0.3 cm could improve the survival rate of the bacteria. The Fourier transform infrared spectroscopy results indicated the interaction between the whey protein, anthocyanins, and the surface composition of the lactic acid bacteria.

## 1. Introduction

At present, lactic acid bacteria powder can be prepared by vacuum drying, spray drying, freeze-drying, and other methods. Freeze-drying is an effective method for preparing and preserving biological materials and has been widely used in obtaining lactic acid bacteria powder. The basic principle of the freeze-drying method is bacteria mixed with a protective medium, which should be pre-frozen below the co-solvent point. Then, you sublimate the ice crystals in the fungus under a high vacuum state below the pressure of the three-phase point to remove the excess water in the fungus, and obtain a dried fermentation agent with a certain water content [1]. The decreased fermentation activity of lactic acid bacteria after lyophilization is due to the damage and irreparable fact of the integrity and stability of the bacterial cell membrane, including fluidity, permeability, etc., which lead to the loss of metabolic enzymes, the decrease of enzyme activity, and the impairment of bacterial physiological metabolic capacity. This could reduce the fermentation activity of lactic acid bacteria due to damage or even death, affecting the industrial application [2]. It was also found that freeze-drying could cause the unsaturated fatty acids in cell membranes to increase significantly, and the greater degree of freeze-drying damage in some strains was due to the weak regulation ability of cell membrane fatty acids [3]. A freeze-drying protective agent has the ability to increase the survival rate of bacteria. Freeze-drying protective agents may be divided into two types: One includes low molecular weight compounds, such as glucose, maltose, sucrose, and some oligosaccharides, which could enter the cells of lactic acid bacteria and inhibit the formation of ice crystals and slow down the growth of ice crystals [4]. This may reduce the damage caused to the bacteria during freezing. The other are macromolecular substances, such as corn stalks, corn cobs, soybean meal, and soybean molasses, which adhere to the surface of bacteria with hydrophilicity and hydrogen bond formation abilities, forming a stable water molecular layer and preventing the bound water in the bacterial cell membrane from transferring outward, which could protect the structure of the bacterial membrane [5,6]. In the freeze-drying process, different types of protective agents have different protective mechanisms for lactic acid bacteria. Salt protects cellular structure by regulating osmotic pressure and pH. Polymers, such as proteins and polysaccharides, act on cell membranes to prevent bacteria from being exposed to the surrounding environment and improve the resistance of cells to the environment [3,7]. The carbon and nitrogen sources in the medium also affect the types and content of unsaturated fatty acids that make up the cell membrane. The selection of appropriate carbon sources can improve the fluidity of cell membranes, make cells show different degrees of freeze resistance, and increase the survival rate after freeze-drying [8]. The pre-freezing conditions, such as temperature and time, are very important for the freeze-drying of the bacteria. When the pre-freezing temperature is relatively higher and the time is shorter, the sample cannot be completely frozen. If the pre-freezing temperature is too low and the pre-freezing time is too long, this may lead to a reduced survival rate [9]. Furthermore, the survival rate of bacteria increased with an increase in freeze-dried thickness, which was due to the number of bacteria per unit volume of different freeze-dried thicknesses. The number of cells per unit volume increased with the thickness of the material layer the longer the drying time required, the lower the survival rate [10].

Whey protein allows powdered products to form a porous structure after freeze-drying. Freeze-dried products with this porous structure are more easily hydrated. Whey proteins adhere to the bacterial surface through hydrogen bonding, preventing cell surface proteins from being exposed or aggregated, stabilizing membrane surface components, and preventing cell damage. Therefore, whey protein has a good protective effect on bacteria and is used as a protective agent for a variety of lactic acid bacteria [11]. Blueberries contain a large amount of anthocyanins, which could interact with whey protein during fermentation processes and form a protective layer around the bacteria to improve the antifreeze ability of the bacteria at the early stage of freeze-drying. In this experiment, the effects of freeze-drying on the survival rate of *Lactobacillus plantarum* 67, *Lactobacillus paracasei* grx701, *Lactobacillus delbrueckii* 134, and *Streptococcus thermophilus* grx02 were investigated in different media, such as an MRS medium, whey protein, and a blueberry–whey protein mixture medium. The freeze-drying process of mixed fermentation of blueberry–whey protein was optimized from different concentrations of blueberry additions, pre-freezing temperatures, and freeze-drying thicknesses so as to improve the survival rate of mixed fermentation of blueberry–whey protein. The protective mechanism of whey protein blueberry mixed fermentation on bacteria against freezing was discussed, which provided a new idea for improving the resistance of bacteria to freezing and increasing the survival rate of freeze-dried powder.

## 2. Materials and Methods

### 2.1. Materials

The *Lactiplantibacillus plantarum* 67, *Lacticaseibacillus paracasei* grx701, *Lactobacillus delbrueckii* 134, and *Streptococcus thermophilus* grx02 were sourced from the Jiangsu Key Laboratory of Dairy Biotechnology and Safety Control. The whey protein (50% protein) was supplied by Fonterra (Auckland, New Zealand). The blueberries (*Vaccinium angustifolium*) were obtained from farmers in Daxinganling, China, during the 2022 harvest period. MRS broth, MRS solid medium, agar, 0.9% normal saline, 10% sucrose solution, 1 mol/L hydrochloric acid, 1 mol/L sodium hydroxide solution, 4 mol/L sodium hydroxide solution, trehalose, sodium glutamate, glycerol, skim milk powder, ultrapure water, M17 broth for *Streptococcus thermophilus* grx02, soybean peptone 5.0 g/L, peptone 2.5 g/L, casein peptone 2.5 g/L, yeast extract 2.5 g/L, beef extract 5.0 g/L, lactose 10.0 g/L, sodium ascorbate 0.5 g/L, sodium β-glycerophosphate 19.0 g/L, and magnesium sulfate 0.25 g/L were sourced from the company of Sinopharm Chemical Reagent Co., Ltd. in Shanghai, China.

### 2.2. Sample Preparation

The blueberry was selected, cleaned, and poured into a juicer. A small small amount of water was then added and we squeezed the juice, which was screened with a gauze three times. The filtrate was added with water, and the blueberry and water were mixed to a 16% proportion to make blueberry juice. The juicer was from the vvmax nutrition center (Super-TNC, Shanghai, China). The fruit and vegetable juice mode was used on the juicer. The mixed sample of blueberry and whey protein was 100 mL, with 6% (*w*/*v*) sucrose, 6% (*w*/*v*) whey protein, and 9% (*v*/*v*) blueberry juice hydrated at 800 rmp and 35 °C for 20 min. The pH of the whey protein concentrate and blueberry mixture was adjusted to 6.5 and 7.0 with food-grade sodium hydroxide, respectively. The sample was heated to a central temperature of 95 °C in a water bath and sterilized for 10 min. A total of 1.5 mL of *L. plantarum* 67 and *L. paracasei* grx 701 (the viable cell number was 10^7^ CFU/mL) at a ratio of 1:1 were inoculated into 100 mL samples (MRS, whey protein solution, and blueberry and whey protein mixed solution), which were cultured at a pH of 6.5 and 37 °C for 12 h. A total of 1.5 mL of *Lactobacillus delbrueckii* 134 and *Streptococcus thermophilus* grx02 at a ratio of 1:1 were inoculated into 100 mL of samples at a pH 7.0, which were cultured at 42 °C for 12 h. All examinations were repeated three times. The three experiments were performed for separate preparations. 

### 2.3. Freeze-Drying Condition

After fermentation, the samples were added to a lyophilized protective agent at a ratio of 2:1 and mixed evenly. The formula of the freeze-dried protectant is 12% skim milk, 4% trehalose, 4% glycerin, and 1.5% sodium glutamate mixed with distilled water. The freeze-dried thicknesses were 0.5 cm, 0.3 cm, and 0.1 cm and were pre-frozen at −20 °C and −80 °C for 24 h, respectively. After which, the viable bacteria were counted and then freeze-dried in a vacuum freeze-dryer for 48 h. The subsequent survival rate of LAB was then calculated.

### 2.4. Determination of Bacterial Survival Rate

The freeze-dried bacterial powder was diluted 10 times with 0.9% normal saline at a gradient of 10^−1^ and shaken at 37 °C for 30 min to completely dissolve the bacterial powder. We diluted the same operation to 10^−2^. A 1 mL diluent was cultured in MRS solid medium for 48 h, and then the concentration of viable bacteria was calculated.

The bacterial survival rate was calculated by the following formula:$$R=\frac{n_{1}}{n_{2}}\times 100\%$$

R is the survival rate of bacteria (%), *n*_1_ is the number of viable lactic acid bacteria after treatment (CFU/mL), and *n*_2_ is the number of viable lactic acid bacteria before treatment (CFU/mL). 

The a_w_ of all the freeze-dried powder samples at 25 °C were measured by a calibrated a_w_ meter (Aqualab, Meter Group, Inc., Pullman, WA, USA) to confirm the equilibrium. All examinations were repeated three times. Collected data are expressed as the mean ± standard deviation (SD).

### 2.5. Fourier Transform Infrared Spectroscopy (FTIR) Analysis

The freeze-dried powder was placed on the operating table. FT-IR spectra were recorded in the ATR mode on a Varian Cary 610/670 FTIR spectrometer (Varian, Salt Lake City, NV, USA) using the Turbo mode of the Ever Glo infrared source. 

### 2.6. SEM Analysis

A small amount of powder was taken from the samples and adhered to the specimen stage. A layer of gold film was coated on the surface of the sample with platinum for 150 s. Finally, the scanning electron microscope (GeminiSEM300, Carl Zeiss Ltd., London, UK) was used to observe the samples at a voltage of 20 KV, and a magnification of 200× and 5000×.

## 3. Results and Discussion

### 3.1. The Survival of Bacteria at Different Fermented Condition

Changes in the viable bacteria count and survival rate of *Lactobacillus plantarum* 67 and *Lactobacillus paracasei* grx701, *Lactobacillus delbrueckii* 134, and *Streptococcus thermophilus* grx02 after growing in different medium and freeze-drying were investigated. The medium was MRS, whey protein, whey protein and blueberry mixture, respectively. The concentration of viable bacteria before and after whey protein fermentation alone and the addition of blueberry juice mixed with *L. plantarum* 67 and *L. paracasei* grx 701 were also investigated to compare the difference in the strain diversity in the whey protein and blueberry medium. The same was also investigated with *Lactobacillus delbrueckii* 134 and *Streptococcus thermophilus* grx02 fermentation processes after freeze-drying.

As shown in Figure 1a, under the same conditions, when the concentration of blueberry juice was 9%, the live bacterial concentrations of *Lactobacillus plantarum* 67 and *Lactobacillus paracasei* grx 701, as well as *Lactobacillus delbrueckii* 134 and *Streptococcus thermophilus* grx02, in the mixed fermented samples of whey protein blueberry were at a maximum before freeze-drying. In the pure blueberry fermentation system, the number of viable bacteria was the lowest. Furthermore, the concentrations of viable bacteria in blueberry fermentation broth alone and whey protein fermentation broth alone were lower than those in the MRS medium. In the blueberry–whey protein mixed fermentation sample, when the concentration of blueberry juice was 9%, the concentration of viable bacteria was the highest. In the mixed fermentation samples of *Lactobacillus delbrueckii* 134 and *Streptococcus thermophilus* grx02, the number of viable bacteria in the pure blueberry fermentation system was less than that in the MRS medium. The concentration of viable bacteria in blueberry and whey protein mixed fermentation solution was also higher than that in the pure whey protein fermentation solution and the MRS medium, and the concentration of viable bacteria first increased and then decreased with the increase in blueberry proportions. It indicated that the mixed fermentation samples of *L. plantarum* 67 and *L. paracasei* grx 701 and the mixed fermentation samples of *Lactobacillus delbrueckii* 134 and *Streptococcus thermophilus* grx02 were most suitable for cell growth when the concentration of blueberry was 9%. As shown in Figure 1b, under the same conditions, when the concentration of blueberry juice was 9% among the blueberry–whey protein mixed samples, the freeze-drying survival rate was the highest in the samples fermented with *Lactobacillus plantarum* 67 and *Lactobacillus paracasei* grx 701 and the samples fermented with *Lactobacillus delbrueckii* 134 mixed with *Streptococcus thermophilus* grx02. When the concentration of blueberry juice increased, the survival rate of lactic acid bacteria decreased, which was related to the antibacterial ability of blueberry polyphenols [12]. Moreover, polyphenols have a greater effect on bacteria [13]. Yue Wu et al. found that the higher viable counts in blueberry and blackberry juices fermented by *B. bifidum* and *L. plantarum* may be associated with the higher content of anthocyanins [14]. Therefore, when the concentration of blueberry juice is 9% in a mixed medium of blueberry and whey protein, it can help to improve the freezing resistance of bacteria.

### 3.2. Effect of Pre-Freezing Temperature on the Lactobacilli Activity

The survival conditions of different fermentation samples after pre-freezing were investigated at pre-freezing temperatures of −20 °C and −80 °C. As shown in Figure 2a, the concentration of bacteria under the condition of −20 °C pre-freezing temperature was generally lower than the survival rate at the temperature of −80 °C pre-freezing temperature after 24 h of pre-freezing of the sample under the same other conditions. As shown in Figure 2b, the survival rate of freeze-dried samples after pre-freezing at −20 °C is generally lower than that at −80 °C. The whey protein and blueberry mixed fermentation could increase the survival rate compared to the whey protein-fermented bacterial powder and MRS-fermented bacteria after freeze-drying. The survival rate of *L. plantarum* 67 and *L. paracasei* grx 701 of blueberry and whey protein mixed fermentation samples was higher than that of *Lactobacillus delbrueckii* 134- and *Streptococcus thermophilus* grx02-fermented samples, as shown in Figure 2b,d. This indicated that the addition of blueberry to the fermentation solution could increase the freezing resistance of lactic acid bacteria. Furthermore, the protective ability of whey protein mixed with blueberry juice for different strains was variable [15]. It was found that the pre-freezing temperature of −80 °C was better than the temperature of −20 °C to protect bacteria and reduce the loss of bacteria. It was due to the large amount of reactive oxygen species (ROS) produced by cells during storage. The ROS was higher in the −20 °C group than in the −80 °C group (*p* < 0.05) [16]. This was due to a reduction in the oxidation rate of fatty acids in the cell membrane in the −80 °C group, thereby better maintaining the activity of the cell membrane [17].

### 3.3. Effect of Freeze-Drying Thickness on the Bacterial Activity

Changes in the survival rate of mixed blueberry and whey protein fermented by *L. plantarum* 67 and *L. paracasei* grx 701, *Lactobacillus delbrueckii* 134, and *Streptococcus thermophilus* grx02 with different freeze-drying thickness were investigated before freeze-drying. The thicker the material layer of the bacteria, the greater the concentration of cells per unit volume and the less damage to the unit lactic acid bacteria during the drying process, as shown in Figure 3a. The survival rate was increased with an increase in freeze-drying thickness. The survival rate was the highest when the freeze-dried thickness was 0.5 cm. Therefore, a freeze-dried thickness of 0.5 cm was the optimum thickness, as shown in Figure 3b. However, a thicker freeze-dried thickness means the bacterial material layer was also thicker and there are a greater number of cells per unit volume, which is not conducive to a complete freeze-drying. In the freeze-drying process, the sublimation interface moves continuously with the extension of drying time, and the direction of movement is mainly from the outside to the inside of the material [18]. In the late stage of drying sublimation, the drying speed is relatively slow, the sublimation boundary moves towards the center of the substance, the thickness of the drying zone gradually increases, and the pressure gradient becomes smaller. Therefore, the water vapor diffusion rate becomes slower [19]. The reduction of the drying thickness is also conducive to energy conservation [20]. As shown in Figure 3a,b, the survival rate and concentration of bacteria in the blueberry mixed whey protein system at a freeze-dried thickness of 0.3 cm were not significant (*p* > 0.05). The time required for the drying process should be minimized to save energy and reduce costs. Therefore, the optimum thickness was the freeze-dried thickness of 0.3 cm. During the freezing process, the cell membrane will be damaged and lose activity. Carbohydrates could improve the integrity of the cell membrane [5]. Blueberries contain polysaccharides, such as galactose, mannose, and glucose. Polysaccharides have excellent film-forming properties and can form a dense glass-like structure. When polysaccharides are combined with water molecules, a network gel is formed, which can effectively slow down the diffusion rate of intracellular substances [21]. Some macromolecular fiber substances form a glass substrate to resist the damage of freeze-drying to cells [5]. Therefore, the bacterial survival rate of the sample added with whey protein and blueberry juice was higher than that of the sample added with whey protein alone. In addition, the co-culture of probiotic strains showed better growth and fermentation abilities. The blueberry juice fermented by lactic acid bacteria could have increased the concentration of phenols but grew better in the co-culture than in the single-strain culture, and the total polyphenol concentration was higher [22]. Proteins and phenols add new space and functional components to form aggregates [23]. Phenolic compounds interact with amino acids through hydrogen bonds [24]. Li Yinghui et al., speculated that the self-aggregation structure of proteins and polyphenols can enhance the tolerance of bacterial cells and protect lactic acid bacteria cells [23]. Therefore, the survival rate of blueberry addition with whey protein was higher than that of the whey protein alone system.

### 3.4. The Water Activity of Freeze-Dried Powders

Moisture content and water activity are different concepts. Water activity (A_w_) is the ratio of the vapor pressure of water in a sample to that of pure water at the same temperature [4]. Moisture content refers to the total content of water in food, which is often expressed by mass fraction. Water activity indicates the state of water in food, which reflects the degree of binding or dissociation between water molecules and food components. In the same kind of food, the higher the water content, the greater the water activity, but for different kinds of food, even if the water content is the same, often the water activity is also different. The study investigated the changes in water content and water activity of the mixed fermentation of different bacteria in MRS, whey protein, blueberry juice, and mixed media after freeze-drying.

As shown in Figure 4, the moisture content of the freeze-dried powder of blueberry juice and whey protein mixed fermentation was slightly higher than that of the MRS medium and the pure whey protein medium. This phenomenon was attributed to the blueberry, which lead to a difficulty in removing the water crystals in cells. The active ingredients in blueberries can retain cell structure in freeze-dried samples [25]. The cell water in the freeze-dried powder containing blueberry fermentation will be blocked in the cell tissue [26]. In addition, sugar-rich substances are also not conducive to drying, because low molecular weight sugars will have a huge chemical affinity with the water molecules contained within the juice. Blueberries contain low molecular weight sugars [27]; therefore, due to the chemical affinity, it is more conducive to the loss of water. Therefore, the freeze-dried powder containing blueberry fermentation products has a higher water content. It was found that *Lactobacillus casei* and *Lactobacillus plantarum* grew better in co-cultures than in single-strain cultures [22]. The relative loose binding of *Lactobacillus plantarum* cells increase the ability of cell surface water absorption [5]. However, the interaction between sugars and *Lactobacillus delbrueckii* reduced water binding sites in the solid state [28]. *Lactobacillus delbrueckii* and *Streptococcus thermophilus* are not tolerant to their environment. The bacterial activity was reduced with the pH value of the fermentation system [29]. Therefore, the water activity of *Lactobacillus delbrueckii* 134 and *Streptococcus thermophilus* grx02 fermentation fermented freeze-dried powder is low. According to the literature, a value below 0.6 is the ideal value to avoid microbial growth, and it is stable between 0.20 and 0.40 [30]. The water activity of *L. plantarum* 67- and *L. paracasei* grx 701-mixed whey protein and blueberry juice freeze-dried powder was between 0.2 and 0.4, and the water activity of *Lactobacillus delbrueckii* 134 and *Streptococcus thermophilus* grx02 fermentation fermented whey protein or blueberry juice freeze-dried powder was less than 0.2. Many hydroxyl groups of polyphenols replace water molecules in the freeze-drying stage, maintaining their original natural structure of cellular biomacromolecules in the hydrated state. Some macromolecular fibers form a protective layer around the bacteria resisting the damage of cells [5].

### 3.5. Analysis of Blueberry–Whey Protein Mixed Fermentation Bacteria by FTIR

The FTIR was used to evaluate the surface structure changes in the freeze-dried powder, which was *L. plantarum* 67- and *L. paracasei* grx 701-fermented whey protein or blueberry juice under different conditions and the same as the strains of *Lactobacillus delbrueckii* 134 and *Streptococcus thermophilus* grx02. As shown in Figure 5a, the amide I band related to C=O and C=N stretching exists in all samples in the range of 1700–1600 cm^−1^. The amide II band contributed to C-N stretching coupled with N-H bending mode exists at 1500–1400 cm^−1^. The wave number of 1300–1200 cm^−1^ was attributed to amide III (C-N stretching and N-H deformation). Wave number ranges of 1700–1600 cm^−1^ and 1500–1400 cm^−1^ indicate secondary structural changes in whey protein [31]. As shown in Figure 5b, the infrared spectrum of fermented blueberries shows a sharp absorption peak at 3500–3100 cm^−1^, which is caused by the stretching and bending vibration of O-H groups. The peak appears in the wavelength range of 3632–3200 cm^−1^, which was contributed to -CH stretching. The stretching vibration at 900~950 cm^−1^ was also contributed to the O-H groups from the blueberry juice. At the same time, as shown in Figure 5d, a special peak was generated at the wavelength of 950–1100 cm^−1^ due to the stretching of the C-O-C group. The peaks of 850–750 cm^−1^ represent the -CH extension from the aromatic amino acid ring of whey protein [32]. The intensity of the amide I and amide II bands in the fermentation samples of MRS medium decreased slightly, and the intensity of the amide I and amide II bands in the fermentation samples of whey protein alone also decreased slightly compared to the mixed fermentation of blueberry–whey protein. This indicated that the content of aromatic ring compounds in the samples fermented by *L. plantarum* 67 and *L. paracasei* grx 701 MRS medium and whey protein medium alone were lower than that of blueberry–whey protein mixed fermentation samples. Aromatic compounds can interact with whey protein. For example, it can spontaneously interact with whey protein and casein through electrostatic interaction, hydrogen bonding, and hydrophobic interaction, resulting in the quenching of protein fluorescence, changing the protein structure. Therefore, adding blueberry in the fermentation system of *L. plantarum* 67 and *L. paracasei* grx 701 has a good effect on inhibiting the degradation of whey protein. This may also be related to the -OH of polyphenols from blueberry juice increased the intensity at a wave number in the range of 3632–3200 cm^−1^. The stretching of =C-O-C at 1050~1000 cm^−1^ attributed to the oxygen attached to the aromatic ring was increased in the blueberry juice and whey protein fermented by the *L. plantarum* 67 and *L. paracasei* grx 701 system, as shown in Figure 5d [33]. The peak at wave number of 950~985 cm^−1^, related to the single substitution reactions of olefins, also decreased with whey protein and blueberry juice in the fermentation system, which indicated an interaction between the whey protein, blueberry, and the surface bond of the strains. The peak of the intermolecular hydrogen bonds at 3570~3450 cm^−1^ decreased when the whey protein and blueberry juice were fermented by the strains of 67 and grx 701 when compared to the black line of *L. plantarum* 67 and *L. paracasei* grx 701 alone [34]. However, the intramolecular hydrogen bonds at ~1400 cm^−1^ were increased compared to the whey protein fermented alone, as shown in Figure 5c. This also indicated the interaction between whey protein and anthocyanins, or the surface constituents of the strain. Furthermore, the hydrogen bonds were attributed to this interaction. The peak of the hydroxyl group of phenols related to C-O stretching vibration was increased at wave number 1100 cm^−1^ with the addition of blueberry juice compared to whey protein fermentation alone [35]. Furthermore, the fermentation of the blueberry mixed with whey protein was compared with the fermentation of whey protein alone. Mixed fermentation weakened the strength of the N-H bond in proteins and the C-O bond in phenols. This also indicated that there was a natural interaction between blueberry and whey protein. However, the vibration groups on the surface of the *Lactobacillus delbrueckii* 134 and *Streptococcus thermophilus* grx02 (red line) was lower than that of *L. plantarum* 67 and *L. paracasei* grx 701 (black line). This was due to the various surface structures of the different strains. The intensity of *Lactobacillus delbrueckii* 134 and *Streptococcus thermophilus* grx02 whey protein powder sharply increased mainly due to the whey protein group. When the blueberry juice mixed with whey protein fermented by *Lactobacillus delbrueckii* 134 and *Streptococcus thermophilus* grx02 had the lowest intensity in FTIR, which may be attributed to the anthocyanins in blueberries buried inside the freeze-dried powder. As shown in Figure 5a, the intensity of *Lactobacillus delbrueckii* 134 and *Streptococcus thermophilus* grx02 was lower than that of *L. plantarum* 67 and *L. paracasei* grx 701 alone. However, the intensity of whey protein fermented by *Lactobacillus delbrueckii* 134 and *Streptococcus thermophilus* grx02 was higher than the freeze-dried powder of whey fermented by *L. plantarum* 67 and *L. paracasei* grx 701. This indicated that the intermolecular and intramolecular hydrogen bond forces between whey protein fermented by *Lactobacillus delbrueckii* 134 and *Streptococcus thermophilus* grx02 were higher than the stain of *L. plantarum* 67 and *L. paracasei* grx 701. This means that the anthocyanins in blueberries were probably between the whey protein and the surface of the strains, which led to a significant decrease in intensity after blueberries were involved in the fermentation. Therefore, the interaction between blueberry and whey protein was better than the traditional MRS medium and pure whey protein medium on the bacteria protection effect.

### 3.6. Surface Morphology of Freeze-Dried Powder

The bacteria cultured by the MRS medium, whey protein solution, and blueberry-mixed whey protein medium added with the protective agent were freeze-dried and observed by SEM. As shown in Figure 6, the freeze-dried bacterial powder, which was cultured by MRS medium, showed the characteristics of many pores, large and rough under a magnification of 200×. The freeze-dried powder had a lamellar structure and no aggregated or dispersed particles. The freeze-dried powder showed no bareness, smooth surface, and almost no porosity. The freeze-dried bacterial powder, which was cultured by mixed blueberry and whey protein, formed a clustered spherical structure with a loose structure and good particle dispersion, when viewed under a magnification of 5000×. The aggregated spherical structure may have a certain protective effect on the bacteria. For blueberry fermentation samples, the freeze-dried bacterial powder was connected into a sheet structure. For the samples of whey protein fermentation alone, the freeze-dried bacterial powder had a spherical structure, a certain sense of granularity, and dispersion [34]. For the bacteria grown in MRS medium, the freeze-dried bacteria powder mixed with the protective agent showed a flaked appearance and no spherical aggregation, indicating that no external coating was formed in the growth process of the MRS medium. In the mixed fermentation process of blueberry and whey, the bacteria were wrapped in the protein, forming a ball; and, when covered by a protective agent, it also gathered in a granular shape after freeze-drying. This suggests that the mixed fermentation of whey protein and blueberry will increase the coating and protection of bacteria. Therefore, a mixed fermentation of blueberry–whey protein combined with protective agents can increase the ability of lactic acid bacteria to resist the stress of a freeze-drying environment.

## 4. Conclusions

The effects of MRS, whey fermentation alone and blueberry–whey protein mixed fermentation on the viability of bacteria during storage under freeze-drying conditions were compared. The structure changes showed that the anthocyanins, whey protein, and lactic acid bacteria in the sample changed the moisture content and water activity of the bacteria powder during the freeze-drying process. When the blueberry juice content is 9% in the blueberry and whey protein mixture, the freeze-drying survival rate of *L. plantarum* 67 mixed with *L. paracasei* grx 701 and *Lactobacillus delbrueckii* 134 mixed with *Streptococcus thermophilus* grx02 was the highest. When the blueberry juice concentration increased to 12% and 15% percent, the survival rate decreased with the increase in added blueberry juice, which was due to the bacteriostatic ability of polyphenols. The survival rate and viable cell count in different mediums at the pre-freezing temperature of −80 °C were higher than those at −20 °C. Moreover, the survival rate of the strains was also higher after pre-freezing at −80 °C than at −20 °C. Though the survival rate of bacteria in MRS and whey protein medium was not significant (*p* > 0.05) compared to the whey protein and blueberry juice mixed medium, the viable cell count was significantly lower than in the whey protein and blueberry juice mixed medium (*p* < 0.05). The optimum freeze-drying thickness was 0.3 cm. The FTIR results indicated that the surface groups of the strains, proteins, and anthocyanins from blueberries interacted with each other during the fermentation process. This structural interaction could protect the strains from low-temperature damage. However, this interaction was related to the stain’s surface structure. In the future, we will further investigate specific information on the interaction of whey proteins, anthocyanins, and the surface components of lactic acid bacteria.

## Figures and Tables

**Figure 1 foods-13-02260-f001:**
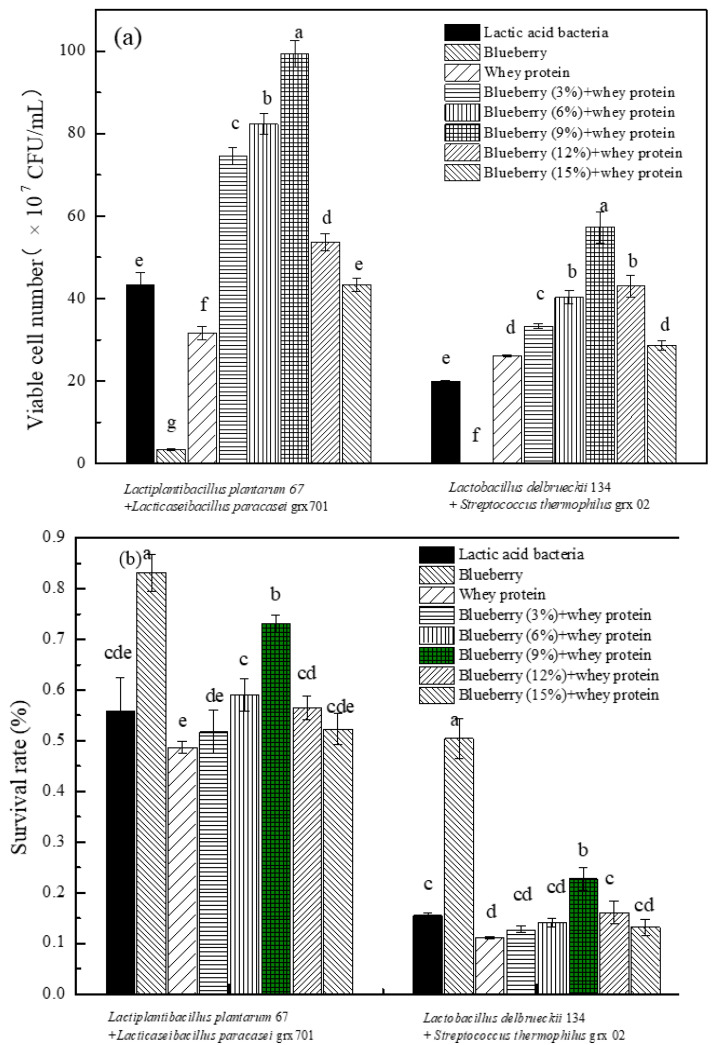
The concentration of viable bacteria (**a**) and survival rate (**b**) of mixed fermentation with different concentrations of blueberry juice addition sample after freeze-drying. Small letters show Lactic acid bacteria represent the mixture *L. plantarum* 67 and *L. paracasei* grx701 or *Lactobacillus delbrueckii* 134 and *Streptococcus thermophilus* grx02 in MRS medium; the blueberry represents blueberry fermented by the mixed lactic acid bacteria; the whey protein represents whey protein fermented by the mixed lactic acid bacteria; and the blueberry + whey protein represents the mixed blueberry and whey protein fermented by the mixed lactic acid bacteria.

**Figure 2 foods-13-02260-f002:**
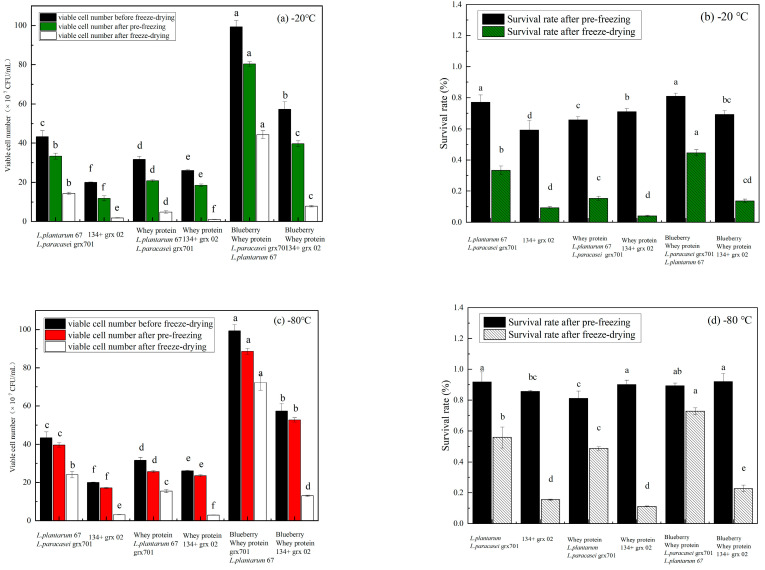
Viable bacteria counts and survival rate using pre-freezing temperatures of −20 °C (**a**,**c**) and −80 °C (**b**,**d**). Small letters show the mixture *L. plantarum* 67 and *L. paracasei* grx701 or 134 and grx02 represent the bacteria cultured in MRS medium; The whey protein + mixed lactic acid bacteria represents the bacteria cultured in whey protein; The blueberry + whey protein + mixed lactic acid bacteria represents the bacteria cultured in mixed blueberry and whey protein system; The blueberry + mixed lactic acid bacteria represents the bacteria cultured in blueberry.

**Figure 3 foods-13-02260-f003:**
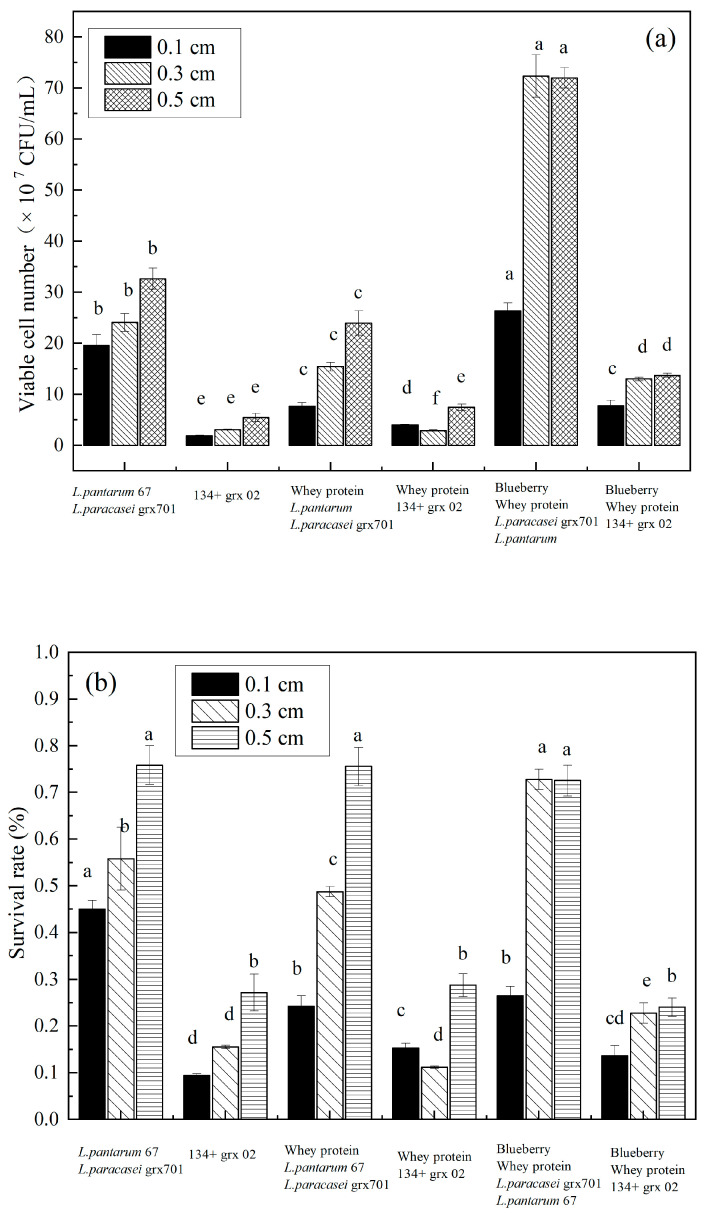
The concentration of viable bacteria (**a**) and survival rate (**b**) at different lyophilized thicknesses during freeze-drying process. Small letters show the mixture *L. plantarum* 67 and *L. paracasei* grx701 or 134 and grx02 represent the bacteria cultured in MRS medium; The whey protein + mixed lactic acid bacteria represents the bacteria cultured in whey protein; The blueberry + whey protein + mixed lactic acid bacteria represents the bacteria cultured in mixed blueberry and whey protein system; The blueberry + mixed lactic acid bacteria represents the bacteria cultured in blueberry.

**Figure 4 foods-13-02260-f004:**
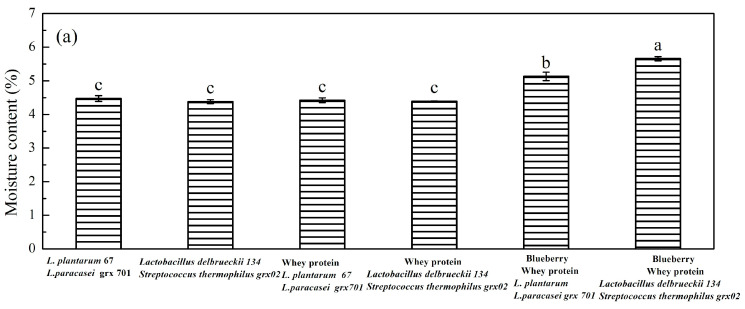
Moisture content (**a**) and water activity (**b**) of *L. plantarum* 67- and *L. paracasei* grx 701-, *Lactobacillus delbrueckii* 134-, and *Streptococcus thermophilus* grx02-fermented whey protein or blueberry juice freeze-dried powder. Small letters show the mixture *L. plantarum* 67 and *L. paracasei* grx701 or 134 and grx02 represent the bacteria cultured in MRS medium; The whey protein + mixed lactic acid bacteria represents the bacteria cultured in whey protein; The blueberry + whey protein + mixed lactic acid bacteria represents the bacteria cultured in mixed blueberry and whey protein system; The blueberry + mixed lactic acid bacteria represents the bacteria cultured in blueberry.

**Figure 5 foods-13-02260-f005:**
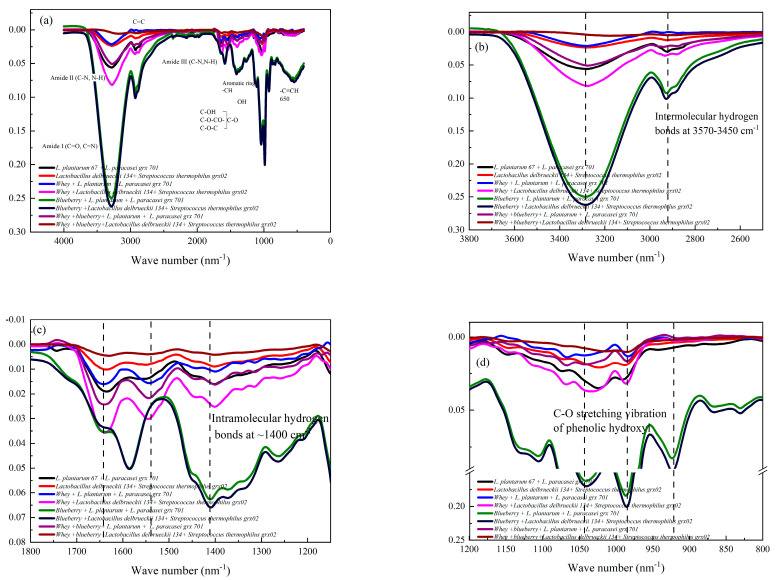
Fourier transform infrared spectrum of *L. plantarum* 67, *L. paracasei* grx 701, *Lactobacillus delbrueckii* 134, *Streptococcus thermophilus* grx02 fermentation fermented whey protein or blueberry juice (**a**). The infrared spectra were analyzed at 3600–2800 cm^−1^ (**b**), 1700–1200 cm^−1^ (**c**), and 1200–950 cm^−1^ (**d**).

**Figure 6 foods-13-02260-f006:**
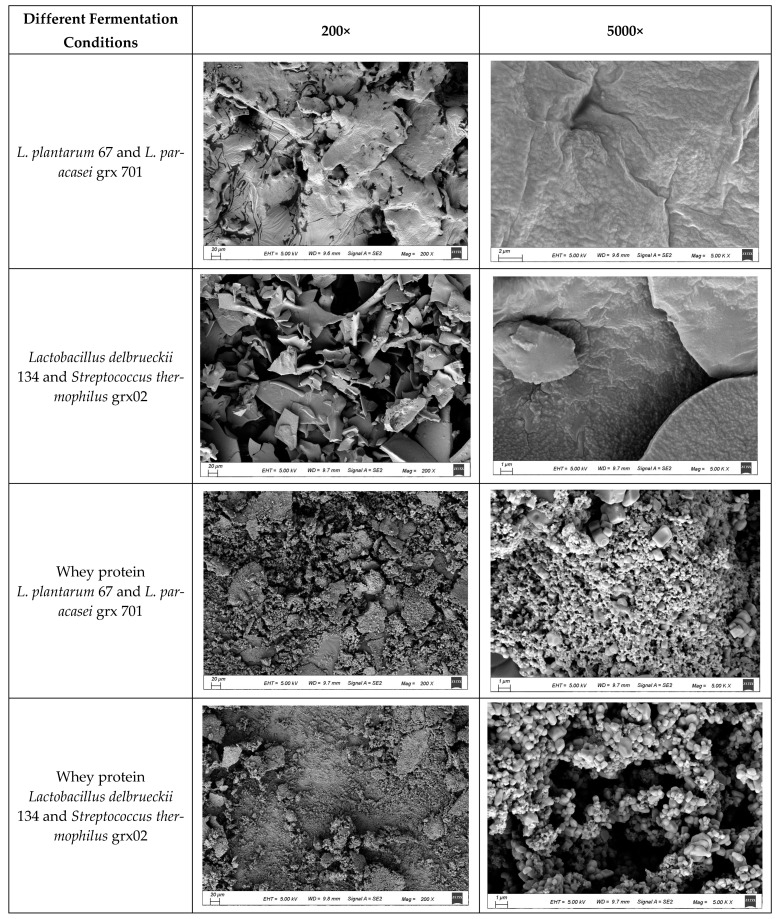
Surface morphology of freeze-dried bacteria powder under different conditions.

## Data Availability

The original contributions presented in the study are included in the article, further inquiries can be directed to the corresponding author.

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
