# Peer review of "Cheese Whey Protein and Blueberry Juice Mixed Fermentation Enhance the Freeze-Resistance of Lactic Acid Bacteria in the Freeze-Drying Process"

_foods, 2024, doi:10.3390/foods13142260_

Round 1

Reviewer 1 Report

Comments and Suggestions for Authors

  The manuscript describes a study aimed at investigating whether cheese whey protein and blueberry juice can improve the resistance of lactic acid bacteria to freeze drying. The objectives are interesting for their application potential. However, it is very difficult to read and evaluate it well because the English used needs a lot of revision. Only after adapting it, the scientific content can be adequately evaluated. Some observations about English are presented below. Other observations are now indicated:  

- Introduction: summarize, it is very hard to read

- 2.2. At this point (review the entire text) when talking about the strains, genus and species are written but not how the strains used are identified (grx701, 134.....). If I write only genus and species I am saying that all the strains of this genus and species give the same result, which is not the case. Only certain strains were used in the study

- lines 132 and 134: how long were they cultivated?

- 2.1: What was the M17 broth used for? Rewrite entire point 2.1

- line 149: were all counts done on MRS agar medium? At what temperature were the plates incubated? How was the count done for S. thermophilus?

- throughout the text we talk about "number of viable bacteria". If I write CFU/ml or g, I am not talking about "numbers" but rather a concentration, which is different. Correct all text and figure titles.

- Figure 1: it is necessary to indicate in the Fig., which part of it is a) and which b)

- line 224: replace "lactobacillus" with "lactobacilli"

- line 245: what is ROS?

- Figure 2: the title must be changed. Write "Viable bacteria counts and survival rate using pree-freezing temperatures ob -20 C (a,c) and - 80 C (b,d)"

- Fig. 4: improve the title (fermentation fermented ??)

- Fig. 6: it is necessary to indicate what working conditions the first lines correspond to

- Results and Discussion: summarize a little, it's hard to read

Comments on the Quality of English Language

- Title: "lactic acid bacteria"

- In References, the genus and species of microorganisms and fruits/plants must be written in italics

- Check all the English of the text. There are many phrases that are not understood. As an example, I cite lines 22-24, 31-32, 46, 56-60, 65-66, 69-70, 136-137............ -

Reviewer 2 Report

Comments and Suggestions for Authors

The manuscript reported of an extensively study on the use of blueberry juice to improve the freeze-resistance of some LAB in the freeze-drying process.

Some minor revisions are needed:

- par 2.3: Authors repeated the pre-freeze conditions twice. 

- Fig.1: the viable bacteria number and survival rate seems the highest for pure blueberry juice and not for the mixing ratio of blueberry and whey protein as stated by the authors. Insert the scale on y axix for the number of viable bacteria.

- P4 L182-183: This sentence ws not not correct for pure blueberry juice results as stated before

-P4 L189-190:  This sentence was not correct for pure blueberry juice. Repetitions of the same concepts as  already stated before

Author Response

On behalf of my co-authors, we thank you very much for giving us an opportunity to revise our manuscript, we appreciate editor and reviewers very much for their positive and constructive comments and suggestions on our manuscript entitled “Cheese whey protein and blueberry juice mixed fermentation enhance the freeze-resistance of Lactic acid bacteria in the freeze-drying processg”. (Manuscript ID: foods-3085009).

We have studied reviewers’ comments carefully and have tried our best to revise our manuscript according to the comments. Those comments are all valuable and very helpful for revising and improving our paper, as well as the important guiding significance to our researches. We have studied comments carefully and have made correction which we hope meet with approval. Revised portion are marked in red in the paper.

The main corrections in the paper and the responds to the reviewer’s comments are as flowing:

Comments and Suggestions for Authors

The manuscript reported of an extensively study on the use of blueberry juice to improve the freeze-resistance of some LAB in the freeze-drying process.

Some minor revisions are needed:

- par 2.3: Authors repeated the pre-freeze conditions twice.

Answer: Thank you very much. The repeated sentence “The samples were freeze-dried for 48 h, and the viable bacteria were counted. ” was deleted in 2.3.

- Fig.1: the viable bacteria number and survival rate seems the highest for pure blueberry juice and not for the mixing ratio of blueberry and whey protein as stated by the authors. Insert the scale on y axix for the number of viable bacteria.

Answer: Thank you very much. The Figure 1(a) represent the concentration of viable bacteria was not inserted in the manuscript. Now the Figure 1(a) was inserted in the manuscript. Figure 1(b) represent the survival of the bacteria.

- P4 L182-183: This sentence ws not not correct for pure blueberry juice results as stated before

Answer: Thank you very much. The sentence “Furthermore, the concentration of viable bacteria in pure blueberry and pure whey protein fermentation broth was less than that in MRS liquid medium. ” was changed into “Furthermore, the concentration of viable bacteria in blueberry fermentation broth alone and whey protein fermentation broth alone was lower than that in MRS Liquid medium.” in line 159-162..

-P4 L189-190: This sentence was not correct for pure blueberry juice. Repetitions of the same concepts as already stated before

Answer: Thank you very much. The sentence “The concentration of viable bacteria in the mixed fermentation broth of blueberry and whey protein was higher than that of pure whey protein fermentation broth and MRS liquid medium, and the concentration of viable bacteria increased and then decreased with the increasing proportion of blueberry.” was changed into “The concentration of viable bacteria in the mixed fermentation broth of blueberry and whey protein was higher than that in whey protein fermentation broth and MRS liquid medium, and the concentration of viable bacteria increased and then decreased with the increasing proportion of blueberry.” in line 165-168.

Reviewer 3 Report

Comments and Suggestions for Authors

In the submitted article, the effects of MRS, whey protein, and blueberry alone and mixed fermentation on the survival rate of lactic acid bacteria-Lactiplantibacillus plantarum, Lacticaseibacillus paracasei, Lactobacillus delbrueckii and Streptococcus thermophilus under various freeze-drying conditions were investigated. The topic addresses an essential aspect of food microbiology and biotechnology, and using natural additives like blueberry juice and whey protein is innovative. Introduction is informative, but moderate editing of the English language is required. In the material and method sections, more details about the experiment should be provided. The results are well presented, however more statisticall analysis would be benficial. In the conclusions, some future directions of the research should be mentioned.

Specific comments:

Page 1, avoid using abbreviations without explanation in the abstract.

Lines 31-32, 46, 61,93 check English for clarity and accuracy.

116, replace "our laboratory" with the specific name or location of the laboratory.

Lines 121-122, avoid using imperative when explaining sample preparation.

Lines 123-124, be more specific about the proportions of added water

Line 125, ratio or precent is intended, please clarify?

Page 3, be more specific about devices, state the type and the producer.

Page 4, line 164, be more precise about the amount of powder used for SEM.

When mentioning Figures in the text, do not use the abbreviation Fig.

Page 9, Figure 4, uses a small a for water activity.

Page 10, compare FTIR results with the existing literature and provide more detailed discussion.

Page 13, provide more detailed discussion for surface morphology of freeze-dried bacteria powder under different conditions.

Line 456, ratio or precent intended, please clarify?

Comments on the Quality of English Language

Lines 31-32, 46, 61,93 check English for clarity and accuracy.

Author Response

On behalf of my co-authors, we thank you very much for giving us an opportunity to revise our manuscript, we appreciate editor and reviewers very much for their positive and constructive comments and suggestions on our manuscript entitled “Cheese whey protein and blueberry juice mixed fermentation enhance the freeze-resistance of Lactic acid bacteria in the freeze-drying processg”. (Manuscript ID: foods-3085009).

We have studied reviewers’ comments carefully and have tried our best to revise our manuscript according to the comments. Those comments are all valuable and very helpful for revising and improving our paper, as well as the important guiding significance to our researches. We have studied comments carefully and have made correction which we hope meet with approval. Revised portion are marked in red in the paper.

The main corrections in the paper and the responds to the reviewer’s comments are as flowing:

In the submitted article, the effects of MRS, whey protein, and blueberry alone and mixed fermentation on the survival rate of lactic acid bacteria-Lactiplantibacillus plantarum, Lacticaseibacillus paracasei, Lactobacillus delbrueckii and Streptococcus thermophilus under various freeze-drying conditions were investigated. The topic addresses an essential aspect of food microbiology and biotechnology, and using natural additives like blueberry juice and whey protein is innovative. Introduction is informative, but moderate editing of the English language is required. In the material and method sections, more details about the experiment should be provided. The results are well presented, however more statisticall analysis would be benficial. In the conclusions, some future directions of the research should be mentioned.

Specific comments:

Page 1, avoid using abbreviations without explanation in the abstract.

Answer: Thank you very much. The abbreviations in the abstract was changed into full name. FTIR was changed into Fourier Transform infrared spectroscopy.

Lines 31-32, 46, 61,93 check English for clarity and accuracy.

Answer: Thank you very much. By the time we received your comments, we had already made changes based on the reviewer's comments in the part of introduction.

116, replace "our laboratory" with the specific name or location of the laboratory.

Answer: Thank you very much. The "our laboratory" was changed into “Jiangsu Key Laboratory of Dairy Biotechnology and Safety Control” in line 99.

Lines 121-122, avoid using imperative when explaining sample preparation.

Answer: Thank you very much. The sentence “Take out the selected and cleaned blueberry and pour it into the juicer, add a small amount of water and squeeze the juice which was screened with gauze three times.” was changed into “The blueberry was selected and cleaned and poured it into the juicer, added a small amount of water and squeeze the juice which was screened with gauze three times.”

Lines 123-124, be more specific about the proportions of added water

Answer: The filtrate was added with water, and the blueberry and water were mixed to a 16 % proportion to make blueberry juice.

Line 125, ratio or precent is intended, please clarify?

Answer: Thank you very much. The ratio or precent was inserted “The mixed sample of blueberry and whey protein was 100 mL, with 6% (w/v) sucrose, 6%(w/v) whey protein and 9% (v/v) blueberry juice hydrated at 800 rmp and 35℃ for 20 min.”

Page 3, be more specific about devices, state the type and the producer.

Answer: The sentence “FT-IR spectra were recorded in ATR mode on a Varian Cary 610/670 FTIR spectrometer (Varian, Salt Lake City, NV, USA), using the Turbo mode of the Ever Glo infrared source. ” was inserted 2.5.

2.6: Finally, the scanning electron microscope (GeminiSEM300, Carl Zeiss Ltd, England) was used to observe the voltage of 20 KV, and the magnification was 200X, 5000 X.

Page 4, line 164, be more precise about the amount of powder used for SEM.

Answer: The powder used for SEM is blown onto the table surface and cannot be measured statistically.

When mentioning Figures in the text, do not use the abbreviation Fig.

Answer: Thank you very much. Fig. Has been changed into Figure.

Page 9, Figure 4, uses a small a for water activity.

Answer: Aw has been changed into aw in Figure 4.

Page 10, compare FTIR results with the existing literature and provide more detailed discussion.

Answer: The existing literature has been inserted.

  1. Wang W-Q, Sheng H-B, Zhou J-Y, et al. The Effect of a Variable Initial Ph on the Structure and Rheological Properties of Whey Protein and Monosaccharide Gelation Via the Maillard Reaction[J]. International Dairy Journal, 2021, 113 104896. https://doi.org/10.1016/j.idairyj.2020.104896
  2. Gedik O, Karahan A G. Properties and Stability of Lactiplantibacillus Plantarum Ab6-25 and Saccharomyces Boulardii T8-3c Single and Double-Layered Microcapsules Containing Na-Alginate and/or Demineralized Whey Powder with Lactobionic Acid[J]. International Journal of Biological Macromolecules, 2024, 271 132406. https://doi.org/10.1016/j.ijbiomac.2024.132406
  3. 32. Wen-Qiong W, Jie-Long Z, Qian Y, et al. Structural and Compositional Changes of Whey Protein and Blueberry Juice Fermented Using Lactobacillus Plantarum or Lactobacillus Casei During Fermentation[J]. RSC Advances, 2021, 11 (42): 26291-26302.https://doi.org/10.1039/D1RA04140A
  4. Amiri S, Teymorlouei M J, Bari M R, et al. Development of Lactobacillus Acidophilus La5-Loaded Whey Protein Isolate/Lactose Bionanocomposite Powder by Electrospraying: A Strategy for Entrapment[J]. Food Bioscience, 2021, 43 101222. https://doi.org/10.1016/j.fbio.2021.101222
  5. Zang Z, Tian J, Chou S, et al. Investigation on the Interaction Mechanisms for Stability of Preheated Whey Protein Isolate with Anthocyanins from Blueberry[J]. International Journal of Biological Macromolecules, 2024, 255 127880. https://doi.org/10.1016/j.ijbiomac.2023.127880

Page 13, provide more detailed discussion for surface morphology of freeze-dried bacteria powder under different conditions.

Answer: The sentence “After freeze-drying, the mixture of whey protein and blueberry fermented samples showed loose structure and good particle dispersion. For blueberry fermentation samples, the freeze-dried bacterial powder was connected into a sheet structure.For the samples of whey protein fermentation alone, the freeze-dried bacterial powder had a spherical structure, a certain sense of granularity and dispersion.

Therefore, mixed fermentation of blueberry whey protein combined with protective agents can increase the ability of lactic acid bacteria to resist the stress of freeze-drying environment.” were inserted in figure 6.

Line 456, ratio or precent intended, please clarify?

Answer:The “percent” has been inserted line 416.

Reviewer 4 Report

Comments and Suggestions for Authors

Foods-3085009

 Cheese whey protein and blueberry juice mixed fermentation enhance the freeze-resistance of Lactic acid bacteria in the freeze-drying process.

By Wang Yuxian et al.

Encapsulation of bioactive components and micro-organisms is an authentic and vibrant research field. The authors attempted to improve the survival rate of lactic acid bacteria in freeze-dried preparations with the aid of the antioxidants in blueberry juice and the water binding capacity of whey proteins. Specific combinations of these substances and adapted freezing and freeze-drying conditions resulted in protection of the bacteria.

In general, the data would be of interest to other researchers. There are, however, shortcomings in the research methods and data interpretation which need substantial attention. Linguistic editing is also advised.

Linguistic editing was carried out on pages 1 and 2 to point out examples. There are many more errors throughout the document.

Line 14. Write out the meaning of MRS. The next mention of this is in line 103 with no explanation in-between.

23: …that the interaction… The sentence seems to be incomplete as the syntax makes no sense.

31: …is an effective method for…

32: …to obtain lactic…

32-38: the authors jump between bacteria and fungus. Why?

43: …during the drying process…

49: …in the cell membrane…

51: …agents have the…

52: …agents may be divided in two types.

58: …with their strong…

62: …mechanisms of action…

66: …constituent unsaturated.

67: …choosing a suitable…

69: …the ? living cells… The sentence sems uncomplete. Should it be “the survivability of” ?

75: …long it may lead to a reduction of survival rate.

88: …as lactic acid…

95: …gluconate, ascorbic acid and… …and micro-factors.

96: …-drying process mainly…

100: …storage condition.

113. Materials and methods. How many times was the experiment repeated and how? Was it one experiment divided in three (or more)? Was it three (or more) separate preparations?

The methods for measurement of wateractivity was not described.

114-119: Provide names of manufacturers of chemicals and milk powder.

120-122: write in the past tense.

121: What were the maceration conditions of the juicer? Name of manufacturer and name of model?

124. White sugar? Was this sucrose? Crystals or the solution described in line 117?

129: …and blueberry juice…

130: How was this heated? Directly or in a water bath? How was the temperature measured? What was the up-coming time? How long was the solution held at 95 deg C? How was the solution sterilized after heating at 95 deg C?

Bacteria were inoculated; what was the cell count? Inoculated into pH 7 samples; what sample?

The description of the method is not precise.

141: What was the “freeze-dried protective agent”?

142: …pre-frozen…

160: The freeze-dried powder…

163: SEM. Elaborate on the method of preparation eg. Nature of “surface table” or studs. Name of microscope?

Fig 2. Spelling error in labels: Freeze-drying, not Freeze-dring.

Fig 5d was cut at absorbance 0.15. In so doing it is not clearwhether the two lower lines display an absorbance peak at approx. 980nm only or also a smaller one at approx. 1075nm similar to all the lines above.

How many repeats of spectra were obtained and how were the presented ones selected?

436: Microscopy results should be under a new heading.

How many photos were taken and how were the presented ones selected?

The descriptions of what is to be seen are unclear. They are not based on any visible aspects or facts provided by the literature and are therefore speculation.

No single or loose bacteria are visible in the photos as described; only a dense matrix. It is not clear why some mixtures form a solid matrix while others form granules.

438: The linguistic expression is wrong; The 500X magnification cannot “form a protective film”. Instead: the film may become visible at 5000X magnification.

454: Regarding “the distribution of lactic acid bacteria in the sample”; The results and photos show no evidence of matrixes with a greater or lesser extend of distribution of bacteria. The SEM should have been a good method to show this but did not.

170-355: No support by or link to literature. The few references provided are not linked to the current work by proper comparison.

361-355: No support by or link to literature.

Comments on the Quality of English Language

/

Author Response

On behalf of my co-authors, we thank you very much for giving us an opportunity to revise our manuscript, we appreciate editor and reviewers very much for their positive and constructive comments and suggestions on our manuscript entitled “Cheese whey protein and blueberry juice mixed fermentation enhance the freeze-resistance of Lactic acid bacteria in the freeze-drying processg”. (Manuscript ID: foods-3085009).

We have studied reviewers’ comments carefully and have tried our best to revise our manuscript according to the comments. Those comments are all valuable and very helpful for revising and improving our paper, as well as the important guiding significance to our researches. We have studied comments carefully and have made correction which we hope meet with approval. Revised portion are marked in red in the paper.

The main corrections in the paper and the responds to the reviewer’s comments are as flowing:

Foods-3085009

 Cheese whey protein and blueberry juice mixed fermentation enhance the freeze-resistance of Lactic acid bacteria in the freeze-drying process.

By Wang Yuxian et al.

Encapsulation of bioactive components and micro-organisms is an authentic and vibrant research field. The authors attempted to improve the survival rate of lactic acid bacteria in freeze-dried preparations with the aid of the antioxidants in blueberry juice and the water binding capacity of whey proteins. Specific combinations of these substances and adapted freezing and freeze-drying conditions resulted in protection of the bacteria.

In general, the data would be of interest to other researchers. There are, however, shortcomings in the research methods and data interpretation which need substantial attention. Linguistic editing is also advised.

Linguistic editing was carried out on pages 1 and 2 to point out examples. There are many more errors throughout the document.

Answer: The the language was also revised thought the manuscript and marked in red.

Line 14. Write out the meaning of MRS. The next mention of this is in line 103 with no explanation in-between.

23: …that the interaction… The sentence seems to be incomplete as the syntax makes no sense.

31: …is an effective method for…

32: …to obtain lactic…

32-38: the authors jump between bacteria and fungus. Why?

43: …during the drying process…

49: …in the cell membrane…

51: …agents have the…

52: …agents may be divided in two types.

58: …with their strong…

62: …mechanisms of action…

66: …constituent unsaturated.

67: …choosing a suitable…

69: …the ? living cells… The sentence sems uncomplete. Should it be “the survivability of” ?

75: …long it may lead to a reduction of survival rate.

88: …as lactic acid…

95: …gluconate, ascorbic acid and… …and micro-factors.

96: …-drying process mainly…

100: …storage condition.

Answer: Thank you very much. By the time we received your comments, we had already made changes based on the reviewer 1 comments in the part of introduction and rewrite. All the above problems have been corrected.

 Materials and methods. How many times was the experiment repeated and how? Was it one experiment divided in three (or more)? Was it three (or more) separate preparations?

The methods for measurement of water activity was not described.

Answer: According to the reviewer’s advice, the content of  The aw of all the freeze-dried powder samples at 25 °C were measured by a calibrated aw meter (Aqualab, Meter Group, Inc., Pullman, WA) to confirm the equilibrium. All examinations were repeated three times. Collected data are expressed as the mean ± standard deviation (SD). has been inserted in 2.4.  The experiment repeated was separate preparations.

114-119: Provide names of manufacturers of chemicals and milk powder.

Answer: According to the reviewer’s advice, the content of 2.1 was changed into "The Lactiplantibacillus plantarum 67, Lacticaseibacillus paracasei grx701, Lactobacillus delbrueckii 134, Streptococcus thermophilus grx02 were from Jiangsu Key Laboratory of Dairy Biotechnology and Safety Control. Whey protein (50% protein) was supplied by Fonterra (Auckland, New Zealand). Blueberries (Vaccinium angustifolium) were obtained from farmers in Daxinganling in China during the 2022 harvest period. MRS broth, MRS solid medium, agar, 0.9 % normal saline, 10 % sucrose solution, 1 mol/L hydrochloric acid, 1 mol/L sodium hydroxide solution, 4 mol/L sodium hydroxide solution, trehalose, sodium glutamate, glycerol, skim milk powder, ultrapure water. M17 broth for Streptococcus thermophilus grx02, soybean peptone 5.0 g/L, peptone 2.5 g/L, casein peptone 2.5 g/L, yeast extract 2.5 g/L, beef extract 5.0 g/L, lactose 10.0 g/L, sodium ascorbate 0.5 g/L, sodium β-glycerophosphate 19.0 g/L, magnesium sulfate 0.25 g/L was from the company of Sinopharm Chemical Reagent Co.,Ltd. in China. "

120-122: write in the past tense.

Answer: The sentence has been changed into The blueberry was selected and cleaned and poured it into the juicer, added a small amount of water and squeeze the juice which was screened with gauze three times.

121: What were the maceration conditions of the juicer? Name of manufacturer and name of model?

Answer: According to the reviewer’s advice, the information “The juicer was from the vvmax nutrition center (Super-TNC, Shanghai, China). The fruit and vegetable juice mode was used.” was inserted in 2.2.

  1. White sugar? Was this sucrose? Crystals or the solution described in line 117?

Answer: Not white sugar. It is sucrose solution.

129: …and blueberry juice…

Answer: Yes. The sentence has been changed into “The mixed sample of blueberry and whey protein was 100 mL, with 6% (w/v) sucrose, 6%(w/v) whey protein and 9% (v/v) blueberry juice hydrated at 800 rmp and 35℃ for 20 min.”

130: How was this heated? Directly or in a water bath? How was the temperature measured? What was the up-coming time? How long was the solution held at 95 deg C? How was the solution sterilized after heating at 95 deg C?

Bacteria were inoculated; what was the cell count? Inoculated into pH 7 samples; what sample?

The description of the method is not precise.

Answer: According to the reviewer’s advice, the content of 2.2 has been changed intoThe blueberry was selected and cleaned and poured it into the juicer, added a small amount of water and squeeze the juice which was screened with gauze three times. The filtrate was added with water, and the blueberry and water were mixed to a 16 % proportion to make blueberry juice. The mixed sample of blueberry and whey protein was 100 mL, with 6% (w/v) sucrose, 6%(w/v) whey protein and 9% (v/v) blueberry juice hydrated at 800 rmp and 35℃ for 20 min. The pH of the whey protein concentrate and blueberry mixture was adjusted to 6.5 and 7.0 with food-grade sodium hydroxide, respectively. The sample was heated to the central temperature of 95 ℃ in water bath and sterilized for 10 min. 1.5 mL of L. plantarum 67 and L. paracasei grx 701  (the viable cell number was 107 CFU/mL) at a ratio of 1:1 were inoculated into 100 mL of samples (MRS, whey protein solution, blueberry and whey protein mixed solution), which were cultured at a pH of 6.5 and 37 ℃ for 12 h. 1.5 mL of Lactobacillus delbrueckii 134 and Streptococcus thermophilus grx02 at the ratio of 1:1 were inoculated into 100 mL of samples at a pH 7.0, which were cultured at 42 ℃ for 12 h..”

141: What was the “freeze-dried protective agent”?

Answer: In this research, the freeze-dried protective agent was added in different mediums before pre-freezing.

142: …pre-frozen…

Answer: “pre-frozen” has been changed into “pre-freezed”

160: The freeze-dried powder…

Answer:freeze-drying powder has been changed into “freeze-dried powder” in line 136.

163: SEM. Elaborate on the method of preparation eg. Nature of “surface table” or studs. Name of microscope?

Answer: According to the reviewer’s advice, the content of 2.6 was changed into A small amount of powder was taken from the sample and adhered to the specimen stage. A layer of gold film was coated on the surface of the sample with platinum for 150 s. Finally, the scanning electron microscope (GeminiSEM300, Carl Zeiss Ltd, England) was used to observe the voltage of 20 KV, and the magnification was 200X, 5000X. ”

 Fig 2. Spelling error in labels: Freeze-drying, not Freeze-dring.

Answer: Thank you very much. The spelling has been changed into “Freeze-drying”.

Fig 5d was cut at absorbance 0.15. In so doing it is not clearwhether the two lower lines display an absorbance peak at approx. 980nm only or also a smaller one at approx. 1075nm similar to all the lines above.

How many repeats of spectra were obtained and how were the presented ones selected?

Answer: The spectrum is repeated three times, and the lines are cut because of the high absorption intensity, so that all results are clearly presented.

436: Microscopy results should be under a new heading.

How many photos were taken and how were the presented ones selected?

The descriptions of what is to be seen are unclear. They are not based on any visible aspects or facts provided by the literature and are therefore speculation.

No single or loose bacteria are visible in the photos as described; only a dense matrix. It is not clear why some mixtures form a solid matrix while others form granules.

Answer: Three photos were taken of each sample at different sites, and a representative picture was selected to be placed in the results.

According to the reviewer’s advice, the content of 3.6 was changed into “ As shown in Figure. 6, the cell surface of the sample fermented by MRS culture agent showed the characteristics of many pores, large and rough under the magnification of 200 X. Under the magnification of 5000 X, the mixed fermentation of blueberry whey protein also formed a certain protective film for the bacteria, which made the bacteria easy to agglomerate together to play a certain protective role. Under the magnification of 5000 X, the bacteria added with the protective agent were observed. The bacteria cultured in MRS medium added with the protective agent before freeze drying and freeze dried for 48 h. Then the freeze dried powder were observed.under the magnification of 5000 X. The freeze-dried powder has a lamellar structure and no aggregated or dispersed particles. The bacteria were wrapped by the protective agent and showed no bareness, smooth surface and almost no porosity. In general, the bacteria powder showed a full shape and complete structure. After freeze-drying, the mixture of whey protein and blueberry fermented samples showed loose structure and good particle dispersion. For blueberry fermentation samples, the freeze-dried bacterial powder was connected into a sheet structure.For the samples of whey protein fermentation alone, the freeze-dried bacterial powder had a spherical structure, a certain sense of granularity and dispersion [33]. The bacteria grown in MRS Medium were almost naked. Therefore, after mixing the protective agent, the freeze-dried bacterial powder appears flake. In the mixed fermentation process of blueberry and whey, the bacteria are wrapped in the protein, forming a ball, and when covered by a protective agent, it will also be gathered in a granular shape after freeze-drying. This suggests that the mixed fermentation of whey protein and blueberry will increase the coating and protection of bacteria.Therefore, mixed fermentation of blueberry whey protein combined with protective agents can increase the ability of lactic acid bacteria to resist the stress of freeze-drying environment.”

438: The linguistic expression is wrong; The 500X magnification cannot “form a protective film”. Instead: the film may become visible at 5000X magnification.

Answer: Thank you very much. The sentence was changed into “ The bacteria cultured in MRS medium added with the protective agent before freeze drying and freeze dried for 48 h. Then the freeze dried powder were observed.under the magnification of 5000 X. The freeze-dried powder has a lamellar structure and no aggregated or dispersed particles. ”

454: Regarding “the distribution of lactic acid bacteria in the sample”; The results and photos show no evidence of matrixes with a greater or lesser extend of distribution of bacteria. The SEM should have been a good method to show this but did not.

Answer: Thank you very much. The sentence “the distribution of lactic acid bacteria in the sample” was deleted.

170-355: No support by or link to literature. The few references provided are not linked to the current work by proper comparison. 361-355: No support by or link to literature.

Answer: According to the reviewer’s advice, the literature has been inserted in the results and discussion.

  1. Wang W-Q, Sheng H-B, Zhou J-Y, et al. The Effect of a Variable Initial Ph on the Structure and Rheological Properties of Whey Protein and Monosaccharide Gelation Via the Maillard Reaction[J]. International Dairy Journal, 2021, 113 104896. https://doi.org/10.1016/j.idairyj.2020.104896
  2. Gedik O, Karahan A G. Properties and Stability of Lactiplantibacillus Plantarum Ab6-25 and Saccharomyces Boulardii T8-3c Single and Double-Layered Microcapsules Containing Na-Alginate and/or Demineralized Whey Powder with Lactobionic Acid[J]. International Journal of Biological Macromolecules, 2024, 271 132406. https://doi.org/10.1016/j.ijbiomac.2024.132406
  3. 32. Wen-Qiong W, Jie-Long Z, Qian Y, et al. Structural and Compositional Changes of Whey Protein and Blueberry Juice Fermented Using Lactobacillus Plantarum or Lactobacillus Casei During Fermentation[J]. RSC Advances, 2021, 11 (42): 26291-26302.https://doi.org/10.1039/D1RA04140A
  4. Amiri S, Teymorlouei M J, Bari M R, et al. Development of Lactobacillus Acidophilus La5-Loaded Whey Protein Isolate/Lactose Bionanocomposite Powder by Electrospraying: A Strategy for Entrapment[J]. Food Bioscience, 2021, 43 101222. https://doi.org/10.1016/j.fbio.2021.101222
  5. Zang Z, Tian J, Chou S, et al. Investigation on the Interaction Mechanisms for Stability of Preheated Whey Protein Isolate with Anthocyanins from Blueberry[J]. International Journal of Biological Macromolecules, 2024, 255 127880. https://doi.org/10.1016/j.ijbiomac.2023.127880

Round 2

Reviewer 1 Report

Comments and Suggestions for Authors

The authors have corrected most of the observations made. There are some left to adjust and they are detailed:

- line 85: Delbroueri ? (whatever species it is, it is in lowercase and italics)

- lines 85 and 158: thermoophilus ?

- 2.1: the trademark of the MRS and M17 Broth media needs to be indicated. It is not necessary to detail the composition of the M17 if the trademark is written

- 2.4: do not use "number of viable bacteria". If I use CFU/ml, this is a concentration

- Fig. 1: missing insert (b) in the Figure

- use italics for genus and species, and species in lowercase) in lines 456, 478, 532, 481, 537 and 539

Comments on the Quality of English Language

watch up

Author Response

On behalf of my co-authors, we thank you very much for giving us an opportunity to revise our manuscript, we appreciate editor and reviewers very much for their positive and constructive comments and suggestions on our manuscript entitled “Cheese whey protein and blueberry juice mixed fermentation enhance the freeze-resistance of lactic acid bacteria in the freeze-drying process”. (Manuscript ID: foods-3085009).

We have studied reviewers’ comments carefully and have tried our best to revise our manuscript according to the comments. Those comments are all valuable and very helpful for revising and improving our paper, as well as the important guiding significance to our researches. We have studied comments carefully and have made correction which we hope meet with approval. Revised portion are marked in red in the paper.

The main corrections in the paper and the responds to the reviewer’s comments are as flowing:

(Reviewer 1)

The authors have corrected most of the observations made. There are some left to adjust and they are detailed:

- line 85: Delbroueri ? (whatever species it is, it is in lowercase and italics)

Answer: Thank you very much. The content “Lactobacillus Delbroueri ” has been changed into Lactobacillus delbrueckii in line 85.

- lines 85 and 158: thermoophilus ?

Answer: Thank you very much. The content of “Streptococcus thermoophilus was changed into  “Streptococcus thermophilus”.

- 2.1: the trademark of the MRS and M17 Broth media needs to be indicated. It is not necessary to detail the composition of the M17 if the trademark is written

Answer: MRS And M17 media were prepared in the laboratory.

- 2.4: do not use "number of viable bacteria". If I use CFU/ml, this is a concentration

Answer: Thank you very much. The "number of viable bacteria" has been changed into “concentration of viable bacteria”

- Fig. 1: missing insert (b) in the Figure

Answer: Thank you very much. The (b) has been inserted in Figure 1.

- use italics for genus and species, and species in lowercase) in lines 456, 478, 532, 481, 537 and 539

Answer: Thank you very much. The use italics for genus and species, and species in lowercase) in reference has been changed and marked in red.

Reviewer 4 Report

Comments and Suggestions for Authors

Foods-3085009

Cheese whey protein and blueberry juice mixed fermentation enhance the freeze-resistance of lactic acid bacteria in the freeze-drying process.

By Wang Yuxian et al.

This is a second review.

·       The authors have added additional information that was requested by the reviewers.

·       Linguistic editing is still advised. New errors were introduced with the additional information. However, many linguistic errors were not corrected as was advised by the reviewers.

·       Scientific interpretation of the data, specifically the microscopy, is over-stretching the capabilities of the technique.

In the light of the above, the document is not publishable at this stage.

Linguistic editing was carried out on pages 1 and 2 to point out examples. There are many more errors throughout the document.

Line 33. …was pre-frozen…

38-40: Two sentences are connected making the meaning of both incomprehensible

45: 51: …agents have the…  This linguistic error was not corrected.

46: …agents may be divided in two types. This linguistic error was not corrected.

49: …which may reduce the…

66: …long it may lead to…

69: …different freeze-dried thickness. (omit …was different.)

70: …the thickness of…

79: …during fermentation processes…

80: …there are few studies… Also provide references to these few.

86: …different media…

86: What is MRS? This was requested in previous review.

128: …pre-frozen…

146: The freeze-dried…

153: …microscope…was used to observe the voltage… A microscope is not used to observe voltage! Wrong linguistics. Rewrite the sentence.

95.109: Requested in first round of review: Materials and methods. How many times was the experiment repeated and how? Was it one experiment divided in three (or more)? Was it three (or more) separate preparations?

Review request not addressed: Fig 5d was cut at absorbance 0.15. In so doing it is not clearwhether the two lower lines display an absorbance peak at approx. 980nm only or also a smaller one at approx. 1075nm similar to all the lines above.

Review request not addressed: How many repeats of spectra were obtained and how were the presented ones selected?

393: Previous review not addressed; The linguistic expression is wrong; The 500X magnification cannot “form a protective film”. Instead: the film may become visible at 5000X magnification.

395-414: Previous review not addressed; Regarding “the distribution of lactic acid bacteria in the sample”; The results and photos show no evidence of matrixes with a greater or lesser extend of distribution of bacteria. In fact, no bacteria are SEEN or indicated in the photos. The SEM should have been a good method to show this but did not. The authors are over-interpreting the results and are presenting observations and explanations which are not supported by the method and results.

157-310. Previous review not addressed: No support by or link to literature. The few references provided are not linked to the current work by proper comparison. New research should be substantiated by existing literature or shown where it compliments existing literature.

Comments on the Quality of English Language

Proof reading necessary

Author Response

On behalf of my co-authors, we thank you very much for giving us an opportunity to revise our manuscript, we appreciate editor and reviewers very much for their positive and constructive comments and suggestions on our manuscript entitled “Cheese whey protein and blueberry juice mixed fermentation enhance the freeze-resistance of lactic acid bacteria in the freeze-drying process”. (Manuscript ID: foods-3085009).

We have studied reviewers’ comments carefully and have tried our best to revise our manuscript according to the comments. Those comments are all valuable and very helpful for revising and improving our paper, as well as the important guiding significance to our researches. We have studied comments carefully and have made correction which we hope meet with approval. Revised portion are marked in red in the paper.

The main corrections in the paper and the responds to the reviewer’s comments are as flowing:

 (Reviewer 4)

Foods-3085009

This is a second review.

The authors have added additional information that was requested by the reviewers.

Linguistic editing is still advised. New errors were introduced with the additional information. However, many linguistic errors were not corrected as was advised by the reviewers.

Scientific interpretation of the data, specifically the microscopy, is over-stretching the capabilities of the technique.

In the light of the above, the document is not publishable at this stage.

Linguistic editing was carried out on pages 1 and 2 to point out examples. There are many more errors throughout the document.

Line 33. …was pre-frozen…

Answer: Thank you very much. The content “ which was pre-freeze below the co-solvent” was changed into “which was pre-frozen below the co-solvent”.

38-40: Two sentences are connected making the meaning of both incomprehensible

Answer: According to the reviewer’s advice the sentences “The decreased fermentation activity of lactic acid bacteria after lyophilization is due to the integrity and stability of bacterial cell membrane (including fluidity, permeability, etc.) are damaged and irreparablethe fact during the process of lyophilization, which lead to the loss of metabolic enzymes, the decrease of enzyme activity, and the impairment of bacterial physiological metabolic capacity. ” was changed into “The decreased fermentation activity of lactic acid bacteria after lyophilization is due to the damage and irreparablethe fact of integrity and stability of bacterial cell membrane including fluidity, permeability, etc, which lead to the loss of metabolic enzymes, the decrease of enzyme activity, and the impairment of bacterial physiological metabolic capacity. ”. in line 38-40.

45: 51: …agents have the…  This linguistic error was not corrected.

Answer: “Freeze-dried protective agents mainly have two types.” was changed into “ Freeze-drying protective agents mainly have two types.”

The sentence “One is low molecular weight compounds such as glucose, maltose, sucrose and some oligosaccharides,which could enter the cells of lactic acid bacteria and inhibit the formation of ice crystals and slow down the growth of ice crystals [4], which reducing the damage to bacteria during freezing.” was changed into “One is low molecular weight compounds such as glucose, maltose, sucrose and some oligosaccharides,which could enter the cells of lactic acid bacteria and inhibit the formation of ice crystals and slow down the growth of ice crystals [4]. This may reduce the damage of the bacteria during freezing. ”

46: …agents may be divided in two types. This linguistic error was not corrected.

Answer: “Freeze-drying protective agents mainly have two types.” was changed into “Freeze-drying protective agents may be divided in two types.”

49: …which may reduce the…

Answer: The sentence was changed into “This may reduce the damage of the bacteria during freezing.” in line 49.

66: …long it may lead to…

Answer: The sentence “If the pre-freezing temperature is too low and the pre-freezing time is too long leading to the reduced survival rate” was changed into “If the pre-freezing temperature is too low and the pre-freezing time is too long, which may lead to the reduced survival rate.”

69: …different freeze-dried thickness. (omit …was different.)

Answer: The sentence “which was due to the number of bacteria per unit volume of different freeze-dried thickness was different.” was changed into “which was due to the number of bacteria per unit volume of different freeze-dried thickness.”

70: …the thickness of…

Answer: The “thickening” was changed into “thickness”.

79: …during fermentation processes…

Answer: has been changed into “fermentation processes” in line 79.

80: …there are few studies… Also provide references to these few.

Answer:  The sentences “There are fewer studies on the addition of natural nutrients such as blueberries to replace oligofructose, magnesium sulphate, dipotassium hydrogen phosphate, calcium lactate, zinc gluconate, sodium vitamin C and other growth factors and microfactor. ” were deleted.

86: …different media…

Answer: The “medium”was changed into “media” in line 86.

86: What is MRS? This was requested in previous review.

Answer: Thank you very much. The MRS has been changed into “MRS medium”.

128: …pre-frozen…

Answer:The “pre-freezed” was changed into “pre-frozen”.

146: The freeze-dried…

Answer: The “freeze-dryed” was changed into “freeze-dried”.

153: …microscope…was used to observe the voltage… A microscope is not used to observe voltage! Wrong linguistics. Rewrite the sentence.

Answer: Thank you very much. The sentence “Finally, the scanning electron microscope (GeminiSEM300, Carl Zeiss Ltd, England) was used to observe the voltage of 20 KV, and the magnification was 200X, 5000X. ” was changed into “Finally, the scanning electron microscope (GeminiSEM300, Carl Zeiss Ltd, England) was used to observe the samples at the voltage of 20 KV, and the magnification was 200X, 5000X.”

95.109: Requested in first round of review: Materials and methods. How many times was the experiment repeated and how? Was it one experiment divided in three (or more)? Was it three (or more) separate preparations?

Answer: Thank you very much. The sentences “All examinations were repeated three times. The three experiments were separate preparations.” was inserted in 2.2.

Review request not addressed: Fig 5d was cut at absorbance 0.15. In so doing it is not clearwhether the two lower lines display an absorbance peak at approx. 980 nm only or also a smaller one at approx. 1075 nm similar to all the lines above.

Review request not addressed: How many repeats of spectra were obtained and how were the presented ones selected ?

Answer: In Figure 5(a), the total peaks of all the samples were showed without cut. In order to see all lines clearly, the 1200-800 wave number range is enlarged in figure 5(d). The FTIR was detected 3 times. Samples from the same batch were selected in figure 5.

393: Previous review not addressed; The linguistic expression is wrong; The 500X magnification cannot “form a protective film”. Instead: the film may become visible at 5000X magnification.

Answer: Thank you very much. The sentences “As shown in Figure. 6, the cell surface of the sample fermented by MRS culture agent showed the characteristics of many pores, large and rough under the magnification of 200 X. Under the magnification of 5000 X, the mixed fermentation of blueberry whey protein also formed a certain protective film for the bacteria, which made the bacteria easy to agglomerate together to play a certain protective role. Under the magnification of 5000 X, the bacteria added with the protective agent were observed. The bacteria cultured in MRS medium added with the protective agent before freeze drying and freeze dried for 48 h. Then the freeze dried powder were observed.under the magnification of 5000 X.” were changed into “The bacteria cultured by the MRS medium, whey protein solution and blueberry mixed whey protein medium added with the protective agent were freeze dried and observed by SEM. As shown in Figure. 6, the freeze-dried bacterial powder which was cultured by MRS medium showed the characteristics of many pores, large and rough under the magnification of 200 X. The freeze-dried bacterial powder which was cultured by mixed blueberry and whey protein was formed a clustered spherical structure, when it was observed at the magnification of 5000 X. The aggregated spherical structure may have a certain protective effect on the bacteria. ”.

395-414: Previous review not addressed; Regarding “the distribution of lactic acid bacteria in the sample”; The results and photos show no evidence of matrixes with a greater or lesser extend of distribution of bacteria. In fact, no bacteria are SEEN or indicated in the photos. The SEM should have been a good method to show this but did not. The authors are over-interpreting the results and are presenting observations and explanations which are not supported by the method and results.

Answer: Thank you very much. The expression was changed in this part. The specific revision was follow.

“The bacteria cultured by the MRS medium, whey protein solution and blueberry mixed whey protein medium added with the protective agent were freeze dried and observed by SEM. As shown in Figure. 6, the freeze-dried bacterial powder which was cultured by MRS medium showed the characteristics of many pores, large and rough under the magnification of 200 X. The freeze-dried powder has a lamellar structure and no aggregated or dispersed particles. The freeze-dried powder showed no bareness, smooth surface and almost no porosity. The freeze-dried bacterial powder which was cultured by mixed blueberry and whey protein was formed a clustered spherical structure showed loose structure and good particle dispersion, when it was observed at the magnification of 5000 X. The aggregated spherical structure may have a certain protective effect on the bacteria. For blueberry fermentation samples, the freeze-dried bacterial powder was connected into a sheet structure. For the samples of whey protein fermentation alone, the freeze-dried bacterial powder had a spherical structure, a certain sense of granularity and dispersion [33]. For the bacteria grown in MRS Medium, the freeze-dried bacteria powder mixed with the protective agent showed flake appearance, and no spherical aggregation appeared, indicating that no external coating was formed in the growth process of MRS Medium. In the mixed fermentation process of blueberry and whey, the bacteria are wrapped in the protein, forming a ball, and when covered by a protective agent, it will also be gathered in a granular shape after freeze-drying. This suggests that the mixed fermentation of whey protein and blueberry will increase the coating and protection of bacteria. Therefore, mixed fermentation of blueberry whey protein combined with protective agents can increase the ability of lactic acid bacteria to resist the stress of freeze-drying environment.”

157-310. Previous review not addressed: No support by or link to literature. The few references provided are not linked to the current work by proper comparison. New research should be substantiated by existing literature or shown where it compliments existing literature.

Answer: The research about the whey protein or other protein mixed berries replace of MRS medium to protect bacteria during freeze drying was few. According to the reviewer’s advice, the relevant references were inserted in the manuscript. The results of the study were analyzed with reference to the relevant literature. Some of the relevant findings have been inserted into the manuscript. The specific revision was as follow.

Part 3.1

[14]      Wu Y, Li S, Tao Y, et al. Fermentation of Blueberry and Blackberry Juices Using Lactobacillus Plantarum, Streptococcus Thermophilus and Bifidobacterium Bifidum: Growth of Probiotics, Metabolism of Phenolics, Antioxidant Capacity in Vitro and Sensory Evaluation[J]. Food Chemistry, 2021, 348 129083. https://doi.org/10.1016/j.foodchem.2021.129083

Part 3.2

This indicated that the blueberry addition in the fermentation solution could increase the freezing resistance of lactic acid bacteria. Furthermore, the protective ability of whey protein mixed blueberry juice to different strains was various [15]. It was found that the pre-freezing temperature of -80 °C was better than the temperature of -20 ℃ to protect bacteria and reduce the loss of bacteria. It was due to the large amount of reactive oxygen species (ROS) produced by cells during storage. The ROS was more in the -20 ℃ group than -80 ℃ group (p < 0.05) [16]. This was due to oxidation rate of fatty acids in the cell membrane was reduction in the -80 °C group,  thereby better maintaining the activity of the cell membrane [17].

Part 3.3

Blueberries contain polysaccharides, such as galactose, mannose and glucose. Polysaccharides have excellent film-forming properties and can form a dense glass-like structure. When polysaccharides are combined with water molecules, a network gel is formed, which can effectively slow down the diffusion rate of intracellular substances [21]. Some macromolecular fiber substances form a glass substrate to resist the damage of freeze-drying to cells [5]. Therefore, the bacterial survival rate of the sample added with whey protein and blueberry juice was higher than that of the sample added with whey protein alone. In addition, the co-culture of probiotic strains showed better growth and fermentation ability. The blueberry juice fermented by lactic acid bacteria could increased the concentration of phenols, but grew better in the co-culture than in the single strain culture, and the total polyphenol concentration was higher [22]. Proteins and phenols add new space and functional components to form aggregates [23]. Phenolic compounds interact with amino acid through hydrogen bonds [24]. Li Yinghui et al speculated that the self-aggregation structure of proteins and polyphenols can enhance the tolerance of bacterial cells and protect lactic acid bacteria cells [23]. Therefore, the survival rate of blueberry addition with whey protein was higher than whey protein alone system.

Part 3.4

Therefore, the freeze-dried powder containing blueberry fermentation products has high water content. It was found that Lactobacillus casei and Lactobacillus plantarum grew better in co-cultures than in single strain cultures [22]. The relative loose binding of Lactobacillus plantarum cells increase the ability of cell surface water absorption [5]. However, the interaction between sugars and Lactobacillus delbrueckii reduced water binding sites in the solid state [28]. Lactobacillus delbrueckii and Streptococcus thermophilus are not tolerant to the environment. The bacterial activity was reduced with the pH value of fermentation sytem [29]. Therefore, the water activity of Lactobacillus delbrueckii 134 and Streptococcus thermophilus grx02 fermentation fermented freeze-drying powder is low. According to the literature, the value below 0.6 is the ideal value to avoid microbial growth, and it is stable between 0.20 and 0.40 [30].